# Roles of Oxidative Stress and Inflammation in Vascular Endothelial Dysfunction-Related Disease

**DOI:** 10.3390/antiox11101958

**Published:** 2022-09-30

**Authors:** Yukihito Higashi

**Affiliations:** 1Department of Regenerative Medicine, Research Institute for Radiation Biology and Medicine, Hiroshima University, Hiroshima 743-8551, Japan; yhigashi@hiroshima-u.ac.jp; Tel.: +81-82-257-5831; 2Division of Regeneration and Medicine, Medical Center for Translational and Clinical Research, Hiroshima University Hospital, Hiroshima 734-8553, Japan

**Keywords:** oxidative stress, inflammation, endothelial function, atherosclerosis

## Abstract

Oxidative stress and chronic inflammation play an important role in the pathogenesis of atherosclerosis. Atherosclerosis develops as the first step of vascular endothelial dysfunction induced by complex molecular mechanisms. Vascular endothelial dysfunction leads to oxidative stress and inflammation of vessel walls, which in turn enhances vascular endothelial dysfunction. Vascular endothelial dysfunction and vascular wall oxidative stress and chronic inflammation make a vicious cycle that leads to the development of atherosclerosis. Simultaneously capturing and accurately evaluating the association of vascular endothelial function with oxidative stress and inflammation would be useful for elucidating the pathophysiology of atherosclerosis, determining treatment efficacy, and predicting future cardiovascular complications. Intervention in both areas is expected to inhibit the progression of atherosclerosis and prevent cardiovascular complications.

## 1. Introduction

Oxidative stress and inflammation play an important role in the development of cardiovascular diseases (CVD) [1,2,3,4,5]. The sources of production of reactive oxygen species (ROS) in blood vessels include nicotinamide adenine dinucleotide phosphate (NADPH) oxidase, xanthine oxidase, uncoupled endothelial nitric oxide synthase (eNOS), arachidonic acid metabolic pathway, mitochondrial electron transfer system dysfunction, and catecholamine autoxidation [6,7]. Among these, NADPH oxidase is the most important and is involved in the production of ROS in vivo, and it is activated by various cytokines, vasoactive substances, and shear stress [8,9]. It is well known that ROS have opposing effects on cell viability. ROS induce cell proliferation, hypertrophy, migration, and apoptosis [10,11]. Chronic inflammation, which begins as a biological response to vascular endothelial dysfunction, is also thought to be the primary cause of atherosclerosis [3,4,5]. Various factors including oxidative stress, oxidized low-density lipoprotein (LDL), thrombi, and viral and bacterial infections induce acute and chronic inflammatory cell infiltrates, including neutrophils, lymphocytes, and macrophages, which in turn enhance local vascular inflammation by enhancing the production of inflammatory cytokines by the infiltrating inflammatory cells [12,13]. It is thought that in the presence of CVD, chronic inflammation makes a vicious cycle, leading to the maintenance and progression of atherosclerosis. Therefore, evaluation of vascular function, which is the primary point of action of chronic inflammation, is extremely important for elucidating the pathogenesis of CVD, determining therapeutic efficacy, and predicting prognosis. In this review, we would like to outline the effects of oxidative stress and inflammation on vascular endothelial cells and their involvement in the onset, maintenance, and progression of vascular injury via redox-sensitive signaling in vascular smooth muscle cells (VSMCs).

## 2. Vascular Endothelial Function

The vascular endothelium is anatomically located in the innermost layer of blood vessels, and the vascular system, a closed system consisting of the heart, arteries, and veins, is composed entirely of one cell layer. Endothelium cells secrete vasodilators including nitric oxide (NO), prostaglandin I_2_, C-type natriuretic peptide, and endothelium-derived vascular hyperpolarizing factor as well as vasoconstrictors including endothelin, angiotensin II, prostaglandin H_2_, and thromboxane A_2_ [14,15]. Among these bioactive substances, NO plays a very important role in atherosclerosis. NO is produced and secreted from L-arginine, an essential amino acid, by activation of endothelial NO synthase (eNOS) through mechanical stimulation of vascular endothelial cells or binding to receptors by agonists such as acetylcholine, histamine, and bradykinin or by shear stress from blood flow in vascular endothelial cells [16,17]. The secreted gaseous NO is transmitted by diffusion to nearby VSMCs and activates intracellular soluble guanylate cyclase, which increases the amount of cyclic guanosine monophosphate and relaxes vascular smooth muscle [18]. The normal vascular endothelium maintains vascular function and structure by adjusting the balance between vasodilation and vasoconstriction, proliferation and antiproliferation of VSMCs, coagulation and anticoagulation, inflammation and anti-inflammation, and oxidation and antioxidation [19]. If all of the vascular endothelium in the body could be collected, the total weight would be equivalent to that of the liver. If it could be spread over an entire surface, the total area would be equivalent to six tennis courts and if it could be connected in a row, the length would be 100,000 km or two and a half times around the earth [20]. Atherosclerosis develops and progresses the first stage of vascular endothelial dysfunction [20]. Hypertension, diabetes, dyslipidemia, aging, obesity, smoking, lack of exercise, excessive salt intake, and menopause induce vascular endothelial dysfunction [21]. Aging is the largest factor that determines vascular endothelial function [22]. Vascular endothelial function is now recognized as a predictive factor in the development of CVD. A meta-analysis by Lerman et al. [23] confirmed that endothelial function is an independent predictor of cardiovascular complications. Vascular endothelial function can also be viewed as a target for treatment of atherosclerosis. Endothelial dysfunction is not irreversible and can be corrected with appropriate interventions such as antihypertensive medications, antidiabetic agents, lipid-lowering therapy, and lifestyle modification [24,25,26,27,28,29,30]. Improvement in endothelial dysfunction is expected to reduce the incidence of coronary and cerebrovascular events and improve life expectancy in the future. Figure 1 shows the roles of oxidative stress and inflammation in the process of endothelial dysfunction-related disease.

Several methods for measuring vascular endothelial function are currently used [31,32]. These can be broadly classified into methods that evaluate blood flow and vessel diameter by reactive hyperemia after ischemia and methods that evaluate reactivity by administering a bioactive substance. Unfortunately, there is no gold standard method, but at this stage, plethysmography is considered to best reflect vascular endothelial function. In general, this method evaluates vascular endothelial function by measuring changes in blood flow through selective administration of NO-producing stimulants such as acetylcholine, methacholine, bradykinin, histamine, or NO inhibitors to arteries in the limbs. Various vasoactive substances can be used in this method in addition to NO agonists and antagonists, allowing multifaceted evaluation of the dynamics of various bioactive substances produced and secreted by vascular endothelial cells, rather than examining only NO from a single perspective. The ultrasound-based method, flow-mediated vasodilation (FMD), is evaluated by the change in vessel diameter after reactive hyperemia in an extremity. FMD (% change) is calculated by [(maximum vessel diameter after lifting of the frame − baseline vessel diameter)/baseline vessel diameter] × 100. Measurement of peripheral arterial pulse amplitude in fingers is also used to assess endothelial function. The ratio of peripheral arterial pulse amplitude before and after reactive hyperemia (reactive hyperemia index) is calculated. Measurement of biomarkers in blood or urine is the most convenient and non-invasive method, but, unfortunately, there are currently no biomarkers that can be evaluated. However, there are various problems such as the possibility that they do not directly reflect NO production and the accuracy of measurement. These measurements should be considered as adjuncts to endothelial function assessment using FMD or plethysmography. FMD measurement is currently the most widely used method for evaluating vascular endothelial function and is likely to become more widely used in the future. However, it is also true that the method still has many issues to be resolved and many problems to be solved in the future, including the issue of reproducibility. Each method has its own advantages and disadvantages, and it is desirable to improve, refine, and standardize the method or to develop a new concept of a measurement device [32].

## 3. Oxidative Stress

### 3.1. Oxidative Stress and Vascular Injury

Oxidative stress plays an important role in CVD and CV events through various mechanisms including activation of NADPH oxidase, xanthine oxidase and cyclooxygenase, dysfunction of the mitochondrial electron transfer system, uncoupled eNOS, catecholamine autoxidation and failure of the antioxidative system (Figure 2). Circulating LDL enters the subendothelium and undergoes denaturation by oxidative stress to become oxidized LDL [12,13]. Oxidized LDL induces the expression of monocyte migration factor macrophage chemotactic protein 1 (MCP-1) in vascular endothelial cells and also induces the expression of adhesion factors such as intracellular adhesion molecule-1 (ICAM-1) and vascular cell adhesion molecule-1 (VCAM-1), causing peripheral monocytes to adhere to the endothelium and invade the subendothelium [33]. Oxidized LDL also induces the secretion and production of macrophage colony-stimulating factor by vascular endothelial cells, leading to the maturation and differentiation of monocytes into macrophages. Oxidized LDL induces apoptosis in many cells, but it has been observed that a large amount of oxidized LDL accumulates in macrophages, leading to foam cell formation without inducing apoptosis [34]. It is thought that these foam cells release various cytokines, inducing inflammation and oxidative stress in the vascular endothelium. In atherosclerotic lesions, these systems make a vicious cycle that leads to the formation of atheroma and, in the final stage, to CV events. Oxidized LDL has also been reported to promote apoptosis of VSMCs lining the luminal side of the atheroma (thinning of the fibrous membrane), which may contribute to atheroma fragility and instability [35]. Lysophosphatidylcholine produced during oxidative degeneration of LDL impairs the NO synthesis system and contributes to the decrease in NO production [36]. Oxidized LDL is also known to act in thrombus formation by increasing plasminogen activator inhibitor-1 (PAI-1) production and decreasing tissue plasminogen activator (t-PA) production from vascular endothelial cells [37]. Thus, oxidized LDL is widely involved in the initiation, maintenance, and progression of atherosclerosis and atherogenesis. Lectin-like oxidized LDL receptor-1 (LOX-1), a novel scavenger receptor for oxidized LDL, was cloned in vascular endothelial cells [38]. In vitro, LOX-1 expression is known to be induced by various cytokines and shear stress [39]. Interestingly, LOX-1 is rarely expressed in normal vessels but is strongly expressed in endothelial cells, VSMCs and macrophages in atherosclerotic lesions [39,40].

### 3.2. ROS Metabolism and Oxidative Stress

ROS are composed of free radicals such as superoxide anions, hydroxyl radicals, and NO and free radical metabolites such as hydrogen peroxide and peroxynitrite. Free radicals have one or more unpaired electrons and are much more reactive than non-radicals. ROS are usually kept at low concentrations as signaling molecules, but when produced in excess, they can cause a variety of adverse effects. In a healthy cellular environment, the toxicity associated with excess production of ROS is eliminated by the operation of antioxidant systems such as superoxide dismutase, catalase and glutathione peroxidase [41]. Failure of the antioxidant defense system or excessive free-radical production that exceeds the antioxidants can lead to cellular damage, and administration of antioxidants can be protective against cellular damage [42]. Oxidative stress is defined as a state in which the balance between oxidation and anti-oxidation is upset by a failure of the antioxidant defense system or by an excess of ROS production that exceeds that of antioxidants, and the balance is tilted toward a relative excess of ROS. ROS are diverse in terms of their production systems, operability, and chemical reactivity. When two free radicals react, they share an unpaired electron with each other to form free radical metabolites. In addition, radicals and free radical metabolites initiate chain reactions to form new radicals.

### 3.3. Enzymes Related to the Production of ROS

NADPH oxidase, also known as the Nox enzyme group, is the most important enzyme in the vascular ROS-producing system (Figure 2) [43]. In the inactivated state, p47^phox^ and p67^phox^ are located in the cytoplasm, while p22^phox^ and Nox2 are located at the plasma membrane [43]. Upon mechanical stimuli such as angiotensin II, endothelin-1, thrombin, or pressure loading, p47^phox^ is phosphorylated and p67^phox^ is activated. Phox is phosphorylated, and the cytoplasmic component complex is translocated into the cytoplasm, where it assembles its components to generate enzymatic activity as NADPH oxidase [44]. This requires the guanine nucleotide-binding proteins Rac-2 and Rap-1 [43]. Activation of NADPH oxidase results in the production of superoxide from oxygen. Usio-Fukai et al. [45] demonstrated that p22^phox^ is involved in NADPH oxidase-mediated production of ROS in the aorta of hypertensive rats and that it regulates angiotensin II-induced vascular smooth muscle thickening. Recently, Touyz et al. [46] reported that VSMCs obtained from six human resistance vessels expressed Nox2 upon angiotensin II stimulation. Nox1, a homologue of Nox2, is expressed in human aortic VSMCs and Wistar-Kyoto rat celiac artery smooth muscle cells but not in human resistance vessel smooth muscle cells [47]. Nox4, another homologue of Nox2, is present in both cell types. Furthermore, other NADPH oxidase subunits, including p40^phox^, p47^phox^, p67^phox^ and p22^phox^, have been found to be present in human resistance artery VSMCs [47]. NADPH oxidase is a novel enzyme that is involved in the production of ROS in human vascular endothelial cells and VSMCs [48,49]. NADPH oxidase is still unknown, and we are waiting for more information to be accumulated.

### 3.4. Oxidative Stress and Vascular Endothelial Dysfunction

Vascular endothelial dysfunction is the beginning of the development of atherosclerosis, and as atherosclerosis progresses, the vascular endothelium itself is damaged [19,20]. ROS have a very high binding affinity for NO and contribute to NO inactivation [50]. This sequence of events is based on data from animal studies, and there is no evidence that the same processes exist in humans [51,52]. We have therefore developed a new approach to the treatment of renovascular hypertension [53]. In renovascular hypertension, angiotensin II is overproduced, activating NADPH oxidase, which in turn produces ROS. When angioplasty was performed to improve angiotensin II to normal levels in patients with such excessive ROS production, both serum malondialdehyde and urinary 8-hydroxy-2’-deoxyguanosine excretion, indicators of oxidative stress, improved to normal levels. Vascular endothelial function also improved to normal levels postoperatively. In addition, preoperative administration of the antioxidant vitamin C improved endothelial function, whereas postoperative administration of vitamin C had no effect. This confirms the existence of a process leading to renin-angiotensin system (RAS) hyperactivity-NADPH oxidase activation-ROS overproduction-vascular endothelial damage in humans, although this is a unique case of renal vascular hypertension [53].

Recently, it has been shown that extracellular conversion of nicotinamide mononucleotide to nicotinamide riboside plays an important role in restoration of endothelial dysfunction [54]. In addition, the enzyme nicotinamide N-methyltransferase in the endothelium has been demonstrated to protect against oxidative stress-induced endothelial injury [55]. Therefore, it is expected that supplementations of nicotinamide mononucleotide and nicotinamide riboside and an activator of nicotinamide N-methyltransferase will improve endothelial function under the condition of oxidative stress.

It has been shown in in vitro, in vivo and clinical studies that antioxidants activate the NO/eNOS pathway through decreasing in the amount of ROS [56,57,58]. Several lines of evidence have shown that antioxidants including vitamin E [59,60,61,62], vitamin B12 [63], alpha-lipoic acid [62], N-acethlcystein [64,65], flaxseed [66], L-carnitine [67], catechins [68], coenzyme Q10 [69], pycnogenol [70], and chlorogenic acids [71] improve or augment endothelial function in patients with CVD and in subjects who have cardiovascular risk factors as well as healthy subjects. However, not all of the studies showed the beneficial effects of antioxidants on endothelial function [72,73,74,75,76].

When cells are exposed to oxidative stress or electrophiles, they defend themselves by inducing the expression of oxidative stress response genes such as glutathione synthetase and heme oxygenase 1 [77]. In this mechanism of gene expression in response to oxidative stress, it is important for gene expression to be regulated at the transcriptional level through the antioxidant response element or electrophile responsive element, which are located upstream of the gene [78]. Heterodimers of nuclear factor erythroid 2-related factor 2 (Nrf2), a basic leucine zipper-type transcription factor, and small Maf group factors bind to these regulatory sequences and strongly activate gene expression [79]. Indeed, activation of Nrf2 has beneficial effects on the endothelium in human microvascular endothelial cells through an increase in endothelial permeability, enhancement of mitochondrial respiration, decrease in ROS production and suppression of endothelin 1 [80]. However, in a clinical setting, we cannot deny the possibility that Nrf2 activators are not only effective for improvement of endothelial function but may also have harmful effects on endothelial function by a decrease in living cells [80].

### 3.5. Oxidative Stress and Vascular Smooth Muscle Hypertrophy and Remodeling

ROS play an important role in intracellular signal transduction. Superoxide and hydrogen peroxide activate many signals associated with VSMC proliferation, including phosphorylation of mitogen-activated protein kinases (MAPK) such as p38 and extracellular-regulated kinase 5, induction of proto-oncogenes such as c-fos, c-myc, and c-jun, and activation of the activator protein-1 (AP-1) transcription factor [81]. Hydrogen peroxide also plays a role in regulation of the signal transducer and activator of transcription stimulated by platelet-derived growth factor, and it activates Akt, phosphorylates epidermal growth factor receptors, activates tyrosine kinases and tyrosine phosphatases, and activates ras [82]. These signals induce redox-sensitive VSMC proliferation, hypertrophy, and apoptosis, leading to vascular wall thickening and remodeling. ROS are also known to affect the activity of matrix metalloproteinase-2 (MMP-2) and MMP-9, which polymerize hyaluronan and degrade proteoglucans and collagen [83]. Thus, ROS alter the components of the extracellular matrix and the vascular wall itself through these processes [83]. It has also been reported that inhibition of NADPH oxidase activity suppresses angiotensin II-stimulated hypertrophy of VSMCs and vascular remodeling [84], suggesting that ROS are involved in vascular smooth muscle hypertrophy and remodeling under conditions of RAS activation.

### 3.6. Oxidative Stress and Apoptosis

It is known that vascular endothelial cells and VSMCs exposed to excessive ROS undergo apoptosis. This is because ROS induce apoptosis in vascular endothelial cells by promoting the expression of p38MAPK and caspases and suppressing the expression of Bcl-2 and induce apoptosis in VSMCs by overexpressing Bax and Fas and suppressing the expression of Bcl-2 [85,86]. However, appropriate (physiological) amounts of ROS are known to act in an anti-apoptotic manner. Depending on the amount, ROS can be either a pro- or inhibitory signaling stimulus for apoptosis. Apoptosis in VSMCs may be involved in the shedding of endothelial cells themselves and in the inhibition of the neoplastic effect of injured vessels, while apoptosis in VSMCs may be involved in the disruption of atheroma.

## 4. Inflammation

Chronic inflammation, which begins as a biological response to vascular endothelial dysfunction, is thought to be the primary cause of atherosclerosis. Factors such as oxidative stress, oxidized LDL, thrombi, and viral and bacterial infections induce acute and chronic inflammatory cell infiltrates, including neutrophils, lymphocytes, and macrophages, which in turn enhance local vascular inflammation by enhancing the production of inflammatory cytokines by the infiltrating inflammatory cells [3,4,5,87]. The presence of CVD and chronic inflammation makes a vicious cycle, leading to the maintenance and progression of atherosclerosis. Therefore, evaluation of vascular function, which is the primary point of action of chronic inflammation, is extremely important for elucidating the pathogenesis of vascular disease, determining therapeutic efficacy, and predicting prognosis.

ROS may be strongly involved in vascular inflammation. Excessive ROS activate redox transcription factors such as nuclear factor-kappa B (NF-kB) and AP-1 [81], resulting in enhanced expression of adhesion factors, increased chemokines, cytokine production, and monocyte invasion into the vessel wall [88]. Inhibition of NF-kB activity suppresses interlukin-6 (IL-6), VCAM-1 and MCP-1 expression [89]. Many studies have shown that inflammation may play a role in vascular injury [90,91,92,93,94,95,96,97,98,99,100,101,102,103,104,105,106,107,108,109,110,111,112,113,114,115,116,117,118,119,120,121,122,123,124]. The presence of angiotensin II, oxidized LDL, and inflammatory cytokines activates NADPH oxidase, which in turn induces inflammation under the condition of oxidative stress due to excess ROS. Thus, oxidative stress, inflammation, and vascular endothelial dysfunction are thought to be interrelated and play important roles in the development, progression, and maintenance of atherosclerosis as well as in the process leading to vascular endothelial dysfunction. Not only in experimental animal models but also in humans, vascular endothelial dysfunction caused by chronic inflammation seems to play an important role in the pathogenesis of atherosclerosis. Clinical studies have suggested an association between chronic inflammation and vascular function [91,92,93,94,95,96,97,98,99,100,101,102,103,110,111,112,113,114,115,116,117,118,119,120,121,122,123,124].

### 4.1. Helicobacter Pylori Infection

Helicobacter pylori-infected individuals have elevated levels of high-sensitivity C-reactive protein (hsCRP), indicating the presence of systemic chronic inflammation. We examined vascular function in Helicobacter pylori-infected and non-infected young healthy men without possible risk factors for vascular endothelial dysfunction [91]. Helicobacter pylori-infected subjects showed a greater reduction in vascular endothelium-independent vascular function with sublingual nitroglycerine than did non-infected subjects. Endothelium-independent vasodilation responses to sublingual nitroglycerine administration were similar in Helicobacter pylori-infected and uninfected subjects, but Helicobacter pylori-infected subjects had a significantly lower FMD [91]. In addition, Helicobacter pylori-infected subjects had a significantly higher level of hsCRP than that in non-infected subjects, suggesting that vascular endothelial function is selectively impaired in Helicobacter pylori-infected patients as a result of chronic inflammation.

### 4.2. Periodontal Disease

The incidence of periodontal disease in Japan is estimated to be about 70% [96]. Japanese people suffer from a variety of periodontal diseases ranging from mild gingivitis to severe periodontitis. Although it is debatable whether periodontal disease itself is directly related to the development of atherosclerosis, numerous epidemiological studies have revealed that it plays an important role in the development, maintenance, and progression of atherosclerosis [97,98]. Periodontal disease may also be a model for systemic chronic inflammation. Recently, we used plethysmography to measure changes in forearm blood flow in response to the endothelium-dependent vasodilator acetylcholine and the endothelium-independent vasodilator sodium nitroprusside in patients with periodontal disease and subjects without periodontal disease who had no coronary risk factors [92]. The periodontal disease group had higher blood flow than that in the nonperiodontal disease group, suggesting that periodontal disease is a marker of inflammation. The levels of the inflammatory markers hsCRP and IL-6 were significantly higher in the periodontal disease group than in the nonperiodontal disease group. Forearm blood flow response to acetylcholine was significantly decreased in the periodontal disease group. Periodontal treatment reduced hsCRP and IL-6 to levels of healthy controls and significantly increased the response to acetylcholine. Administration of the NO synthase inhibitor NG-monomethyl-L-arginine prevented the enhanced response to acetylcholine after periodontal treatment. Responsiveness to sodium nitroprusside was comparable in the two groups and did not change after periodontal treatment. A similar study was conducted in patients with hypertension and patients with coronary artery disease, and those patients in the periodontal disease group showed increased inflammatory markers and decreased vascular endothelial function, confirming that periodontal disease intervention improves inflammatory markers and vascular endothelial function [92,93]. Periodontal disease itself has been shown to be a significant factor in the response of patients with hypertension and patients with coronary artery disease [92,93]. We found that periodontal disease itself induces endovascular dysfunction in healthy subjects as well as in patients with hypertension and patients with coronary artery disease due to decreased biological activity of NO caused by chronic inflammation and that treatment of periodontal disease improves endovascular dysfunction.

### 4.3. Kawasaki Disease

Kawasaki disease is an acute inflammation that develops in childhood and causes inflammation of blood vessel walls throughout the body, and it is known to cause coronary artery aneurysms and stenosis in a small number of patients [99]. The possibility that not only vascular damage caused by inflammation in the acute phase but also persistent inflammation over a long period of time may contribute to vascular damage cannot be ruled out. Compared to the control group, the group with Kawasaki disease had a similar endothelium-independent vasodilation response to sublingual nitroglycerine but significantly lower FMD [94]. In an examination of vascular function in coronary arteries with aneurysms and stenosis in patients with Kawasaki disease, the response to acetylcholine has been shown to be decreased [95]. Interestingly, patients with Kawasaki disease who had coronary aneurysms also had decreased vascular endothelial function in forearm arteries without apparent aneurysms or stenosis compared with patients without aneurysms [94]. Outside of the acute inflammatory phase, some reports showed that inflammatory markers, including CRP, are not elevated, while others reports showed that hsCRP is elevated even in the chronic phase of the disease. In any case, there is no doubt that vascular damage associated with inflammation exists in Kawasaki disease.

### 4.4. Bürger’s Disease

Bürger’s disease has another name, thromboangiitis obliterans, and the involvement of chronic inflammation in its development, maintenance, and progression is strongly suspected. In patients with Bürger’s disease, it is possible to measure vascular function in peripheral vessels, which are the primary site of the disease. Compared to controls, patients with Bürger’s disease had significantly lower FMD not only in the below-knee arteries with abnormalities such as stenosis but also in the forearm arteries without angiographic morphological abnormalities [100,101]. Endothelium-independent vasodilatation responses to sublingual nitroglycerine were the same in controls and patients with Bürger’s disease. In addition, there was a significant inverse correlation between hsCRP and FMD. Interestingly, the degree of cell migration, an indicator of vascular endothelial progenitor cell function, showed a significant inverse correlation with the level of hsCRP. The correlations of FMD with endothelial progenitor cell number and function suggest that chronic inflammation may contribute to endothelial dysfunction by reducing the number and function of endothelial progenitor cells. These findings suggest that chronic inflammation may contribute to vascular endothelial dysfunction.

### 4.5. Other Acute and Chronic Inflammatory Diseases

Several lines of evidence have shown that other acute and chronic inflammatory diseases are also associated with endothelial dysfunction [102,103,104,105,106,107,108,109,110,111,112,113,114,115,116,117,118,119,120,121,122,123,124], though the mechanisms of endothelial dysfunction in acute and chronic inflammatory diseases remain unclear. Endothelial function is impaired in various acute inflammatory diseases including acute rheumatic fever [104], systemic inflammation induced by influenza A vaccination in patients with human immunodeficiency virus infection [105], pre-eclampsia [106], acute Kawasaki disease [107], severe sepsis [108], and foot-and-mouth disease [109] and in various chronic inflammatory diseases including rheumatoid arthritis [110,111], juvenile idiopathic arthritis [112], systemic sclerosis [113], systemic lupus erythematosus [114], Behçet’s disease [115], rhinosinusitis [116], ankylosing spondylitis [117], multiple sclerosis [118], bowel disease [119], Takayasu disease [120], psoriasis [121], chronic obstructive pulmonary disease [122], pulmonary vasculitis [123], and myocardial inflammation [124].

The results confirm the existence of a process leading from acute and chronic inflammation to vascular endothelial dysfunction in humans, as shown in patients with Helicobacter pylori infection, patients with periodontal disease, patients with Kawasaki disease, patients with Bürger’s disease, and other inflammatory diseases [90,91,92,93,94,95,96,97,98,99,100,101,102,103,104,105,106,107,108,109,110,111,112,113,114,115,116,117,118,119,120,121,122,123,124]. Vascular endothelial function may be considered a marker of vascular inflammation, although various chemical biomarkers exist for inflammation. Interventions for chronic inflammation may improve vascular endothelial damage and prevent cardiovascular complications in the future. Further studies on chronic inflammation and vascular endothelial function, including basic and clinical studies, are needed. Figure 3 shows mechanisms by which inflammation induce endothelial dysfunction-related disease.

## 5. Conclusions

Oxidative stress and chronic inflammation contribute to atherosclerosis by inducing cell proliferation, cell hypertrophy, apoptosis, and Rho kinase activation via activation of intracellular signaling pathways or by directly damaging vascular endothelium. In addition, there is an interaction between oxidative stress and chronic inflammation. Furthermore, it is thought that endothelial dysfunction-related diseases and oxidative stress and chronic inflammation make a vicious cycle that leads to the maintenance and development of atherosclerosis, which in turn leads to cardiovascular events.

## Figures and Tables

**Figure 1 antioxidants-11-01958-f001:**
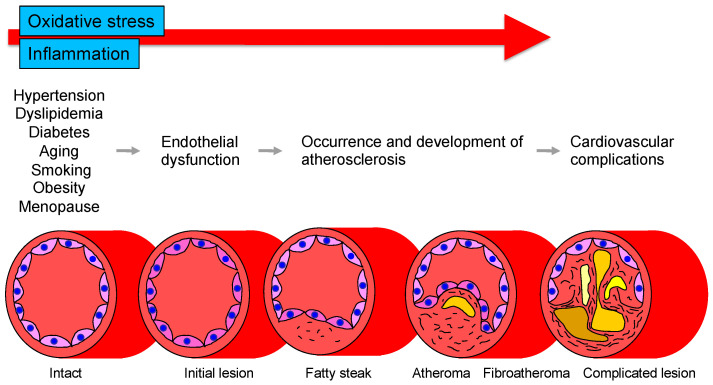
Roles of oxidative stress and inflammation in the process of endothelial dysfunction-related disease.

**Figure 2 antioxidants-11-01958-f002:**
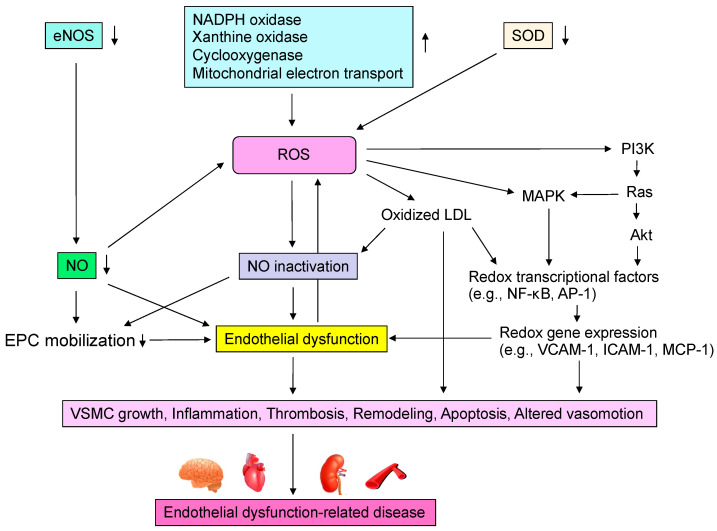
Mechanisms by which reactive oxygen species (ROS) induce endothelial dysfunction-related disease. eNOS, endothelial nitric oxide synthase; NO, nitric oxide; NADPH, nicotinamide adenine dinucleotide phosphate; SOD, superoxide dismutase; LDL, low-density lipoprotein; MAPK, mitogen-activated protein kinase kinases; NF-κB, nuclear factor-kappa B; AP-1, activator protein-1; VCAM-1, vascular cell adhesion molecule-1; ICAM-1, intracellular adhesion molecule-1; MCP-1, macrophage chemotactic protein 1; EPC, endothelial progenitor cell; VSMC, vascular smooth muscle cell.

**Figure 3 antioxidants-11-01958-f003:**
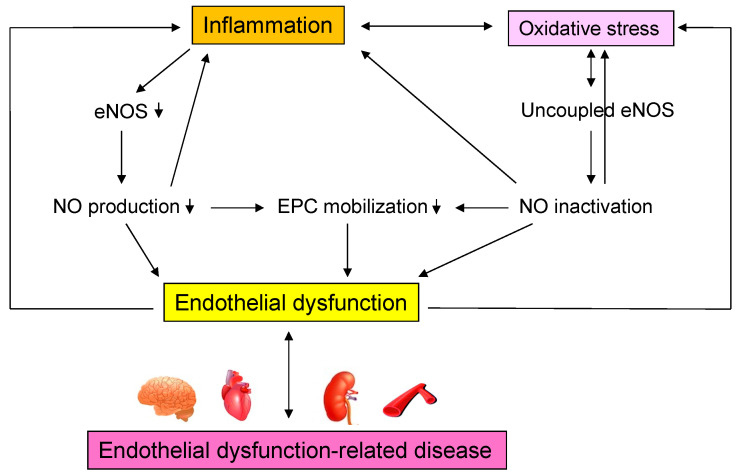
Mechanisms by which inflammation induce endothelial dysfunction-related disease. eNOS, endothelial nitric oxide synthase; NO, nitric oxide; EPC, endothelial progenitor cell.

## Data Availability

Not applicable.

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
