# Peer review of "Roles of Oxidative Stress and Inflammation in Vascular Endothelial Dysfunction-Related Disease"

_antioxidants, 2022, doi:10.3390/antiox11101958_

Round 1
Reviewer 1 Report
The author summarized the current basic and clinical evidence connecting oxidative stress and inflammation with dysfunction of the vascular wall. The topic is of interest, but the manuscript is quite difficult to read in its present form. An overall revision is advisable. In addition, there are some issues that need to be discussed in a more balanced and comprehensive way.
Main comments:
1. Some sentences are very long and too complicated to read. Rephrasing is needed to improve their clarity. For example, page 2 lines, 61-64 or page 3 lines 86-92.
2. Some concepts are repeated within the same paragraph with a different phrasing, as for example page 4 lines 147-160 or page 5 lines 198-204. Other concepts are repeated many times along the manuscript, as for example the enzymes responsible for oxidative stress in CVD (page 1, lines 24-28). A more concise and systematic discussion will improve clarity.
3. The author included opposing effects of ROS on cell viability, as proliferation and apoptosis, in the same sentence (for example page 1 line 31-32 or page 6 lines 228-229). This could generate confusion to the readers and should be explained in a clearer way.
4. In section 2, the ways to measure endothelial function should be described more clearly and the description should include and explain the techniques used in the section 4 (i.e. endothelium-dependent and -independent vasodilation, the use of agents as Ach or nitroglycerine, etc).
5. Paragraph 3.3 is totally focused on NADPH oxidase. Is this supported by the lack of evidence about the involvement of the other enzymatic systems?
6. Paragraph 3.4: the author should consider adding available evidence of the effects of different antioxidants (as vitamins, NAC or others) on endothelial function.
7. Paragraph 3.7: the author should include evidence, if available, of the impact of oxidative stress on the Rho-RACK axis. This is needed to support the direct role of ROS on ROCKs depicted in figure 3. If there are no evidence to date, the importance of this axis for the aim of this present review should be reduced (and likely included in the previous paragraphs); figure 3 should be modified accordingly.
8. Section 4: the selection of the inflammatory conditions is limited to those directly addressed by the author. The discussion should be more balanced including studies on other common acute and chronic inflammatory conditions. I’m not sure the example of Burger’s disease is appropriate in this section.
Minor comments:
9. Consider changing “cardiac” with “coronary” at page 2 line 79.
10. Consider changing “denatured” with “oxidized” or “modified” LDL at page 3. LDL oxidation also occurs in the circulation, not only in the subendothelial space. The description of LOX-1 is not so recent (page 3 line 126).
11. Remove sentences at page 7 lines 281-284.
12. Check the sentence at page 3 lines 114-115.
13. Section 5 is short, and it mainly summarizes previous concepts. The author should consider removing it.
Author Response
Manuscript ID: antioxidants-1932365 R1 Reviewer 1
I would like to thank the reviewer for the helpful comments and hope that we have now produced a more balanced and better account of our work.
The author summarized the current basic and clinical evidence connecting oxidative stress and inflammation with dysfunction of the vascular wall. The topic is of interest, but the manuscript is quite difficult to read in its present form. An overall revision is advisable. In addition, there are some issues that need to be discussed in a more balanced and comprehensive way.
Main comments:
- Some sentences are very long and too complicated to read. Rephrasing is needed to improve their clarity. For example, page 2 lines, 61-64 or page 3 lines 86-92.
Response: In accordance with the reviewer’s suggestion, the sentence “The normal vascular endothelium has vasodilation and contraction, proliferation and antiproliferation of VSMCs, coagulation and anticoagulation, inflammation and anti-inflammation, and oxidation and antioxidant effects, which work in balance to regulate and maintain vascular tone and structure.” has been changed to “The normal vascular endothelium maintains vascular function and structure by adjusting the balance between vasodilation and vasoconstriction, proliferation and antiproliferation of VSMCs, coagulation and anticoagulation, inflammation and anti-inflammation, and oxidation and antioxidation (page 2, lines 61-64).” and the sentences “Accurate measurement of vascular endothelial function is of great clinical importance. Currently, in clinical practice, endothelial function is evaluated by changes in vessel diameter after reactive ischemic hyperemia in the extremities using ultrasound, flow-mediated vasodilation (FMD), by plethysmography, in which various vasoactive substances are directly injected to evaluate NO production stimulants and NO synthesis inhibitors, and by finger vascular endothelial function that is evaluated by measuring biomarkers in blood or urine, although the specificity of this method is lower.” has been changed to “Several methods for measuring vascular endothelial function are currently used. These can be broadly classified into methods that evaluate blood flow and vessel diameter by reactive hyperemia after ischemia and methods that evaluate reactivity by administering a bioactive substance. Unfortunately, there is no gold standard method, but at this stage, plethysmography is considered to best reflect vascular endothelial function. In general, this method evaluates vascular endothelial function by measuring changes in blood flow through selective administration of NO-producing stimulants such as acetylcholine, methacholine, bradykinin, histamine, or NO inhibitors to arteries in the limbs. Various vasoactive substances can be used in this method in addition to NO agonists and antagonists, allowing multifaceted evaluation of the dynamics of various bioactive substances produced and secreted by vascular endothelial cells, rather than examining only NO from a single perspective. The ultrasound-based method, flow-mediated vasodilation (FMD), is evaluated by the change in vessel diameter after reactive hyperemia in an extremity. FMD (% change) is calculated by [(maximum vessel diameter after lifting of the frame - baseline vessel diameter) /baseline vessel diameter] × 100. Measurement of peripheral arterial pulse amplitude in fingers is also used to assess endothelial function. The ratio of peripheral arterial pulse amplitude before and after reactive hyperemia (reactive hyperemia index) is calculated. Measurement of biomarkers in blood or urine is the most convenient and non-invasive method, but, unfortunately, there are currently no biomarkers that can be evaluated. However, there are various problems such as the possibility that they do not directly reflect NO production and the accuracy of measurement. These measurements should be considered as adjuncts to endothelial function assessment using FMD or plethysmography (page 3, lines 86-108).”
- Some concepts are repeated within the same paragraph with a different phrasing, as for example page 4 lines 147-160 or page 5 lines 198-204. Other concepts are repeated many times along the manuscript, as for example the enzymes responsible for oxidative stress in CVD (page 1, lines 24-28). A more concise and systematic discussion will improve clarity.
Response: In accordance with the reviewer’s suggestion, repeated concepts have been removed and have been modified. Therefore, the sentences (page 4, lines 142-144, page 5, 184-193 and 198-204 in the previous version) have been removed.
- The author included opposing effects of ROS on cell viability, as proliferation and apoptosis, in the same sentence (for example page 1 line 31-32 or page 6 lines 228-229). This could generate confusion to the readers and should be explained in a clearer way.
Response: I agree with the reviewer’s comment that ROS have opposing effects on cell viability, as proliferation and apoptosis. Thus, the sentence “It is well known that ROS have opposing effects on cell viability.” has been added before the sentence on page 1, lines 31-32.
- In section 2, the ways to measure endothelial function should be described more clearly and the description should include and explain the techniques used in the section 4 (i.e. endothelium-dependent and -independent vasodilation, the use of agents as Ach or nitroglycerine, etc).
Response: In accordance with the reviewer’s suggestion, I have rewritten the ways to measure endothelial function and the techniques used in Section 2 (page 3, lines 86-108).
- Paragraph 3.3 is totally focused on NADPH oxidase. Is this supported by the lack of evidence about the involvement of the other enzymatic systems?
Response: Yes. Evidence about the involvement of the other enzymatic systems such as xanthine oxidase and unpaired eNOS is scarce. From the publication data, the main enzyme related to the production of ROS is NADPH oxidase. Therefore, I focused on NADPH oxidase in paragraph 3.3.
- Paragraph 3.4: the author should consider adding available evidence of the effects of different antioxidants (as vitamins, NAC or others) on endothelial function.
Response: In accordance with the reviewer’s suggestion, I have added evidence of the effects of different antioxidants (e.g., vitamins, NAC or others) on endothelial function in paragraph 3.4 (lines 221-228). It has been shown in in vitro, in vivo and clinical studies that antioxidants activate the NO/eNOS pathway through decreasing in the amount of ROS [new refs. 56-58]. Several lines of evidence have shown that antioxidants including vitamin E [new refs. 59-62], vitamin B12 [new ref. 63], alpha-lipoic acid [new ref. 62], N-acethlcystein [new refs. 64,65], flaxseed [new ref. 66], L-carnitine [new ref. 67], catechins [new ref. 68], coenzyme Q10 [new ref. 69], pycnogenol [new ref. 70], and chlorogenic acids [new ref. 71] improve or augment endothelial function in patients with CVD and in subjects who have cardiovascular risk factors as well as healthy subjects. However, not all of the studies showed the beneficial effects of antioxidants on endothelial function [new refs. 72-76].
- Paragraph 3.7: the author should include evidence, if available, of the impact of oxidative stress on the Rho-RACK axis. This is needed to support the direct role of ROS on ROCKs depicted in figure 3. If there are no evidence to date, the importance of this axis for the aim of this present review should be reduced (and likely included in the previous paragraphs); figure 3 should be modified accordingly.
Response: As the reviewer pointed out, unfortunately, there are no clear data on the direct impact of oxidative stress on the Rho-ROCK axis. Figure 3 is speculative. Therefore, we have removed Figure 3 and paragraph 3.7.
- Section 4: the selection of the inflammatory conditions is limited to those directly addressed by the author. The discussion should be more balanced including studies on other common acute and chronic inflammatory conditions. I’m not sure the example of Burger’s disease is appropriate in this section.
Response: In accordance with the reviewer’s suggestion, I have added evidence of endothelial dysfunction in other acute and chronic inflammatory diseases. Several lines of evidence have shown that other acute and chronic inflammatory diseases are also associated with endothelial dysfunction [103-125], though the mechanisms of endothelial dysfunction in acute and chronic inflammatory diseases remain unclear. Endothelial function is impaired in various acute inflammatory diseases including acute rheumatic fever [105], systemic inflammation induced by influenza A vaccination in patients with human immunodeficiency virus infection [106], pre-eclampsia [107], acute Kawasaki disease [108], severe sepsis [109], and foot-and-mouth disease [110] and in various chronic inflammatory diseases including rheumatoid arthritis [111,112], juvenile idiopathic arthritis [113], systemic sclerosis [114], systemic lupus erythematosus [115], Behçet's disease [116], rhinosinusitis [117], ankylosing spondylitis [118], multiple sclerosis [119], bowel disease [120], Takayasu disease [121], psoriasis [122], chronic obstructivepulmonary disease [123], pulmonary vasculitis [124], and myocardial inflammation [125]. These comments have been incorporated into Section 4 (lines 383-395). Interestingly, as stated in lines 380-387 of page 9 in the previous version, we found a significant relationship between FMD as an index of endothelial function and hsCRP as an index of inflammation in patients with Burger’s disease, while there were no significant relationships between FMD other confounding factors for endothelial function. Therefore, I would like to keep the description of Burger’s disease in the manuscript.
Minor comments:
- Consider changing “cardiac” with “coronary” at page 2 line 79.
Response: In accordance with the reviewer’s suggestion, “cardiac” has been changed to “coronary”.
- Consider changing “denatured” with “oxidized” or “modified” LDL at page 3. LDL oxidation also occurs in the circulation, not only in the subendothelial space. The description of LOX-1 is not so recent (page 3 line 126).
Response: In accordance with the reviewer’s suggestion, “denatured” has been changed to “oxidized”. The word “Recently” has been deleted in the sentence “Recently, lectin-like oxidized LDL receptor-1 (LOX-1), a novel scavenger receptor for oxidized LDL, was cloned in vascular endothelial cells [38].”
- Remove sentences at page 7 lines 281-284.
Response: In accordance with the reviewer’s suggestion, the sentences on page 7, lines 281-284 has been removed.
- Check the sentence at page 3 lines 114-115.
Response: In accordance with the reviewer’s suggestion, we have changed the sentences “LDL itself is not taken up by macrophages or VSMCs and is only taken up in the form of denatured LDL (oxidized LDL), resulting in foaminess of macrophages (foam cells) of macrophages and vascular smooth muscle [34]. Cocooning of these foam cells in the vessel wall results in the accumulation of cholesterol.” to “Oxidized LDL induces apoptosis in many cells, but it has been observed that a large amount of oxidized LDL accumulates in macrophages, leading to foam cell formation without inducing apoptosis [34]. It is thought that these foam cells release various cytokines, inducing inflammation and oxidative stress in the vascular endothelium.
- Section 5 is short, and it mainly summarizes previous concepts. The author should consider removing it.
Response: In accordance with the reviewer’s suggestion, Section 5 has been removed.
Reviewer 2 Report
The manuscript “Roles of Oxidative Stress and Inflammation in Vascular Endothelial Dysfunction-related Disease” is a review article regarding the impact of oxidative stress and chronic inflammation on the endothelial dysfunction in general and in the pathogenesis of atherosclerosis. I really appreciate the work performed by the author. The review is well written and can be of interest for the readers, only few typos are present. However, in order to le the manuscript be suitable for publication, the following concerns should be addressed:
The review should include some important studies which were ignored in the actual version of the manuscript. For instance, it would be very important to discuss that it can be achieved a reversal of endothelial dysfunction by nicotinamide mononucleotide via extracellular conversion to nicotinamide riboside (PMID: 32389638). Nonetheless, in the paragraph “oxidative stress”, it is recommended to discuss the fact that the enzyme nicotinamide N-methyltransferase in endothelium has been demonstrated to protect against oxidant stress-induced endothelial injury (PMID: 34153425).
Finally, the manuscript would benefit from a discussion relating the effect of agents that are known to activate the antioxidant Nrf2 transcription factor, which however also display side effects (PMID: 33123312).
Author Response
Manuscript ID: antioxidants-1932365 R1 Reviewer 2
I would like to thank the reviewer for the helpful comments and hope that we have now produced a more balanced and better account of our work.
The manuscript “Roles of Oxidative Stress and Inflammation in Vascular Endothelial Dysfunction-related Disease” is a review article regarding the impact of oxidative stress and chronic inflammation on the endothelial dysfunction in general and in the pathogenesis of atherosclerosis. I really appreciate the work performed by the author. The review is well written and can be of interest for the readers, only few typos are present. However, in order to le the manuscript be suitable for publication, the following concerns should be addressed:
The review should include some important studies which were ignored in the actual version of the manuscript. For instance, it would be very important to discuss that it can be achieved a reversal of endothelial dysfunction by nicotinamide mononucleotide via extracellular conversion to nicotinamide riboside (PMID: 32389638). Nonetheless, in the paragraph “oxidative stress”, it is recommended to discuss the fact that the enzyme nicotinamide N-methyltransferase in endothelium has been demonstrated to protect against oxidant stress-induced endothelial injury (PMID: 34153425).
Response: In accordance with the reviewer’s suggestion, we have incorporated comments concerning a reversal of endothelial dysfunction by nicotinamide mononucleotide via extracellular conversion to nicotinamide riboside anddiscussion of the fact that the enzyme nicotinamide N-methyltransferase in the endothelium has been demonstrated to protect against oxidive stress-induced endothelial injury into the oxidative stress section (page 5, lines 214-220). Thus, references (PMID: 32389638 and PMID: 34153425) have been added to the list of references as reference numbers 54 and 55.
Finally, the manuscript would benefit from a discussion relating the effect of agents that are known to activate the antioxidant Nrf2 transcription factor, which however also display side effects (PMID: 33123312).
Response: In accordance with the reviewer’s suggestion, discussion of the effects of agents that are known to activate the antioxidant Nrf2 transcription factor, which however also display side effects, has been incorporated into the oxidative stress section (page 6, lines 229-242). Thus, a reference (PMID: 33123312) has been added to the list of references as reference numbers 80.
Round 2
Reviewer 1 Report
The author revised the paper according to the reviewer's suggestion. Thank you.